# *miR-377* Inhibits Proliferation and Differentiation of Bovine Skeletal Muscle Satellite Cells by Targeting *FHL2*

**DOI:** 10.3390/genes13060947

**Published:** 2022-05-26

**Authors:** Yun Zhu, Peng Li, Xingang Dan, Xiaolong Kang, Yun Ma, Yuangang Shi

**Affiliations:** School of Agriculture, Ningxia University, Helan Mountain West Road 489, Yinchuan 750021, China; zyhsc102085@163.com (Y.Z.); lipeng.3018@163.com (P.L.); danxingang2013@163.com (X.D.); kangxl9527@126.com (X.K.); mayun_666@126.com (Y.M.)

**Keywords:** miR-377, skeletal muscle satellite cells, proliferation, differentiation, wnt/β-catenin signaling, FHL2

## Abstract

Non-coding RNAs, especially microRNAs (miRNAs), play an important role in skeletal muscle growth and development. miR-377 regulates many basic biological processes and plays a key role in tumor cell proliferation, migration and apoptosis. Nevertheless, the function of miR-377 during skeletal muscle development and how it regulates skeletal muscle satellite cells (SMSCs) remains unclear. In the present study, we proposed to elucidate the regulatory mechanism of miR-377 in the proliferation and differentiation of bovine primary SMSCs. Our results showed that miR-377 can significantly inhibit the proliferation of SMSCs. In addition, we found that miR-377 can reduce myotube formation and restrain skeletal myogenic differentiation. Moreover, the results obtained from the biosynthesis and dual luciferase experiments showed that FHL2 was the target gene of miR-377. We further probed the function of FHL2 in muscle development and found that FHL2 silencing significantly suppressed the proliferation and differentiation of SMSCS, which is contrary to the role of miR-377. Furthermore, FHL2 interacts with Dishevelled-2 (Dvl2) to enable Wnt/β-catenin signaling pathway, consequently regulating skeletal muscle development. miR-377 negatively regulates the Wnt/β-catenin signaling pathway by targeting FHL2-mediated Dvl2. Overall, these findings demonstrated that miR-377 regulates the bovine SMSCs proliferation and differentiation by targeting FHL2 and attenuating the Wnt/β-catenin signaling pathway.

## 1. Introduction

Skeletal muscle is the most important component of the carcass of an animal raised for meat, and accounts for 45~60% of its body weight at the adult stage. It plays a key role in maintaining the body’s movement, respiration and metabolism [1,2]. Myogenesis is a strictly controlled program that commences with the commitment of skeletal muscle satellite cells (SMSCs), which are located under the basal lamella that surrounds the muscle fiber, followed by activation and proliferation into myoblasts [3,4]. Skeletal muscle hypertrophy and regeneration after birth mainly depend on SMSCs. The proliferation and differentiation of SMSCs are regulated by many factors, such as genes, signaling pathways, and non-coding RNAs (ncRNAs).

MiRNAs are 22~24 nucleotide ncRNAs that downregulate gene expression levels by inducing the degradation of target mRNAs or by influencing translation by binding to the 3’ UTRs of mRNA. MiRNAs control many basic biological processes such as proliferation, differentiation and apoptosis of SMSCs, and play a core role in skeletal muscle growth and development. Many studies found miR-377 acts as a tumor suppressor by targeting EGR1, SGK3, XIAP ZEB2 CUL4A and others in lung tumorigenesis [5], cervical carcinoma [6], colorectal cancer [7] and ovarian cancer [8]. However, the role of miR-377 in bovine SMSCs and the molecular regulatory mechanism in skeletal muscle remain unclear.

Four-and-a-half-LIM domains (FHL)2 belong to the FHL protein family, which is prevailingly expressed in cardiac and skeletal muscle [9]. Family members contain two highly reserved finger domains, connected with arranged zinc, each of which has four highly reserved cysteine residues that bind a zinc atom [10]. Homozygous deletion of FHL2 in mice causes different forms of cardiomyopathy and heart failure [11,12]. Studies have shown that FHL2 protein is concerned in regulating the expression of tissue-specific genes by combining with disparate transcription factors [13]. Wei et al. found that FHL2 showed increased expression in hepatoblastomas and activated the Wnt signaling pathway as a coactivator of β-catenin [14]. Martin et al. also found that FHL2 combines with β-catenin to promote the differentiation of mice myoblasts [15]. However, the importance of FHL2 as a regulator in the development of bovine skeletal muscle has not been reported. At present, some studies have found that miRNA regulates cell growth and development by targeting FHL2. Algaber et al. found that miR-340-5p can antagonize colon cancer cell metastasis by targeting the FHL2-E-cadherin axis [16]. Kong et al. found that miR-195-5p accelerated the development of ovarian cancer by targeting FHL2 [17]. However, no studies have reported that miRNA regulates skeletal muscle growth and development by targeting FHL2.

In this research, we hypothesized that miR-377 regulates skeletal muscle growth and development. To test this tentatively, we decided the effects of miR-377 mimic or inhibit the proliferation and differentiation of bovine SMSCs in vitro and evaluated the role of FHL2.

## 2. Materials and Methods

### 2.1. Cell Culture

Bovine SMSCs were isolated form longissimus muscle of a 2-week-old Holstein bull calf, as previously described [16]. SMSCs were obtained by percoll digestion and differential centrifugation. SMSCs were expanded in growth medium consisting of Dulbecco’s Modified Eagle Medium (DMEM; Sigma, St. Louis, MO, USA), 10% fetal bovine serum (FBS; Gibco, Grand Island, NY, USA), and 1% antibiotics-antimycotics (Sigma) at 37 °C under 5% CO_2_. To induce differentiation, growth medium was replaced by differentiation medium consisting of DMEM, 2% horse serum (Hyclone, Logan, UT, USA), and antibiotics-antimycotics.

### 2.2. RNA Oligonucleotides, Vectors, and Transfection

The SMSCs were seeded in 6-well plates at a concentration of 0.5 × 10^5^ cells per well on the day before transfection. Each transfection was performed using Lipofectamine 3000 (Invitrogen, Carlsbad, CA, USA) as per the manufacturer’s instructions. Cells were transfected with miR-377 inhibitor, inhibitor negative control (NC), miR-377 mimic and mimic NC. All RNA oligonucleotides are listed in Table 1.

To silence FHL2, bovine satellite cells were stably transfected with a pcDNA3.1 vector containing a FHL2 short hairpin RNA (shRNA) cassette. Stable clones were produced using the transfection reagent Lipofectamine 3000 (Invitrogen) based on the manufacturer’s instructions. Western blotting (WB) and real-time quantitative (qRT)-PCR were used to monitor the expression of FHL2 and conduct further analyses.

### 2.3. RNA Isolation and Real-Time Quantitative PCR (qRT-PCR)

Total RNA was leached with TRIzol reagent (Invitrogen) according to the manufacturer’s instructions. Approximately 2 µg RNA was reverse transcribed using the Takara PrimeScript RT reagent kit (Dalian, Liaoning, China) according to the manufacturer’s instructions. Then, about 10 ng of cDNA equivalents were selected with SYBR for real-time PCR. The qRT-PCR was performed essentially as described previously [17]. Statistical analysis of qRT-PCR results was performed by determining the mean threshold cycle (ΔCt) values for the expression of standardized genes [18]. Details of the primers used are shown in Table 2.

### 2.4. Western Blotting (WB) and Immunoprecipitation (IP) Analysis

The SMSCs were washed with phosphate-buffered saline (PBS) then lysed with RIPA lysis buffer (BestBio, Shanghai, China). The protein concentration was measured with a bicinchoninic acid protein detection kit (BestBio, Shanghai, China). For IP analysis, the cells were lysed with IP lysis buffer and the total cell lysate including proteins was immunoprecipitated with DVL-2 or FHL2 antibodies. The immune complex was washed with IP lysis buffer (Beyotime, Shanghai, China) three times, separated by sodium dodecyl sulfate–polyacrylamide gel electrophoresis (SDS-PAGE), and transferred to a polyvinylidene fluoride membrane for WB analysis, which was performed using standard procedures. Total nuclear proteins of cells were extracted using kits from Bestbio, Shanghai, China according to the manufacturer’s instructions. The protein concentration was resolved using the bicinchoninic acid assay kit. (Vazyme, Nanjing, China). The following antibodies were used: mouse anti-MyHC (cat. no. M4276, Sigma, St. Louis, MO, USA, 1:2000), mouse anti-MyoG (cat. no, sc-12732, Santa Cruz Biotechnology, Santa Cruz, CA, USA, 1:2000), rabbit anti-FHL2 (cat. no, sc-52667, Santa Cruz, 1:1500), rabbit anti-Active-β-catenin (cat. no. 8814s, Cell Signaling Technology, Danvers, MA, USA, 1:1500), rabbit anti-Dvl-2 (cat. no. ab22616, Abcam, Cambridge, UK, 1:1000), rabbit anti-Histone H3 (cat. no. ab1791, Abcam, 1:2000), rabbit anti-β-actin (cat. no. SAB5500001, Sigma, 1:5000) and mouse anti-β-actin (cat. no. A3854, Sigma, 1:5000).

### 2.5. Cell Counting Kit-8 (CCK-8) Assay

The number of living cells at 12, 24, 36 and 48 h after initiation of culture was determined using a non-radioactive CellTiter 96 assay kit (Sigma) according to the manufacturer’s instructions. The absorbance at 450 nm reflects the number of living cells. The cell proliferation rate is expressed as the slope of the line connecting absorbance at different time periods of culture. All the experiments were repeated ten times independently.

### 2.6. 5-Ethynyl-20-Deoxyuridine (EdU) Assay

Cell proliferation of transfected cells was also measured using an EdU assay kit (Invitrogen, Carlsbad, CA, USA). Following the 48 h shRNA transfection, cells were cultured in 24-well plates and incubated with 10 μL of 10 mmol/L EdU for 2 h. After this, the plates were kept in an incubator for 30 min and then fixed with 4% paraformaldehyde for 15 min. After punching holes with 0.5% TritonX-100 at room temperature for 20 min, 0.5 mL of Click-iT reaction cocktail was added to each well and incubated away from light for 30 min. Next, the nuclei were stained with Hoechst. EdU-positive cells in random fields were counted.

### 2.7. Luciferase Reporter Assay

For the luciferase reporter assay, the cells were co-transfected with 200 nM miR-377 mimic or NC (RiboBio Co., Ltd., Guangzhou, China) and 650 ng of FHL2-3′-UTR-WT and FHL2-3′-UTR-MUT. Cells were collected 48 h after transfection and analyzed with the Dual-Luciferase Reporter Assay System (Yeasen Biotechnology Co., Ltd., Shanghai, China). Moreover, the cells were co-transfected with the Basic vector and NC was used as a control.

### 2.8. Statistical Analysis

GraphPad was employed for statistical analysis. Statistical analysis performed in the results of the three technical replicates performed for each sample. Statistically significant differences were calculated using an unpaired two-tailed Student’s t test. A difference was considered significant when the *p* value was < 0.05. All data are expressed as means ± standard error of the mean (SEM).

## 3. Results

### 3.1. Expression Pattern of miR-377 in Bovine

Based on RT-PCR, the results showed that miR-377 is highly expressed in skeletal muscle and heart (Figure 1A), so we conjectured that miR-377 may play an important role in skeletal muscle development. Therefore, we tested the expression of miR-377 in the proliferation and differentiation stages of bovine SMSCs and found that the expression level of miR-377 reduced significantly (Figure 1B). To investigate the function of miR-377 in the proliferation and differentiation of bovine SMSCs, we transfected miR-377 inhibitors and mimics to SMSCs to promote and inhibit its expression levels, respectively (Figure 1C,D).

### 3.2. miR-377 Inhibit the Proliferation of Bovine SMSCs

To investigate the effect of miR-377 on the proliferation of bovine SMSCs, we detected the proliferation status of bovine SMSCs through quantitative Real-time PCR (qPCR), 5-Ethynyl-20-deoxyuridine (EdU) and Cell Counting Kit-8 (CCK8) assays. CCK8 assay found that the proliferation of SMSC cells increased significantly in miR-377 silenced cells (Figure 2A). The cells transfected with miR-377 inhibitor had greater expression levels of CyclinD1, CDK2 and PCNA mRNA than those negatively transfected with the inhibitor (Figure 2B). EDU assay also found that cell proliferation significantly increased in those transfected with miR-377 inhibitor SMSCs (Figure 2C). We achieved mir-377 overexpression with the transfected cell with the miR-377 mimics. However, we found that miR-377 overexpression significantly inhibited the proliferation of bovine SMSCs (Figure 2D–F). Together, these results manifest that miR-377 restrains the proliferation of bovine SMSCs.

### 3.3. miR-377 Inhibits the Differentiation of Bovine SMSCs

To further explore the effect of miR-377 on bovine skeletal muscle development, we further detected the effect of miR-377 on the differentiation of bovine SMSCs. We found that miR-377 knockdown significantly promoted the mRNA expression levels of myogenin (MyoG), myosin heavy chain 3 (MyH3), creatine kinase muscle (Ckm) and myoglobin (Mb) (Figure 3A), which are markers of the myogenic gene. Western blot analysis showed that the protein expressions levels of MyHC and MyoG were also significantly increased in those transfected with miR-377 inhibitor bovine SMSCs (Figure 3B). Immunofluorescence staining of MyHC analysis showed that miR-377 silencing fortified SMSCs differentiation and increased the total area of myotubes (Figure 3E). In contrast, miR-377 overexpression significantly inhibited the differentiation of bovine SMSCs (Figure 3C,D,F). These results suggest that miR-377 negatively regulates the differentiation of bovine SMSCs.

### 3.4. miR-377 Directly Target FHL2 and Inhibit Its Expression

To investigate the mechanism of miR-377 inhibiting proliferation and differentiation of bovine SMSCs, we used TargetScan as a gene prediction program to identify proliferation and differentiation-related genes containing miR-377 response elements in 3′-UTR. The results showed that FHL2 is a candidate target of miR-377, and studies have shown that miR-377 has effects on proliferation and differentiation. In order to confirm this result, a destabilizing mutation was introduced into the predicted binding sites (Figure 4A). Luciferase results showed that exogenous miR-377 reduced luciferase activity in SMSCs cells transfected with 3-UTR FHL2 but had no effect on luciferase activity of mutated FHL2 3-UTR (Figure 4B). We found that both mRNA and protein levels of FHL2 were significantly reduced after transfection with miR-377 mimic, while transfection with miR-377 inhibitor significantly promoted the expression of FHL2 (Figure 4C–E). These results indicate that FHL2 is a validated target of miR-377 in bovine SMSCs.

### 3.5. FHL2 Knockdown Inhibited the Proliferation of SMSCs

To investigate the role of FHL2 during the proliferation of bovine satellite cells, the SMSCs were transfected with control shRNA or FHL2 shRNA and cultured in growth medium. The expression of FHL2 was significantly decreased after transfection with FHL2 shRNA compared with control shRNA, as demonstrated by qPCR (Figure 5A) and Western blotting (Figure 5B). CCK8 assay showed that FHL2 knockdown significantly impeded the proliferation of bovine SMSCs at 24 and 36 h post transfection (Figure 5C). The mRNA expression of proliferation marker genes CyclinD1, CDK2 and PCNA were significantly decreased after 24 h of FHL2 silencing, as shown by qPCR (Figure 5D). EdU staining manifested that the proliferation rate of cells under FHL2 silencing was significantly decreased compared with control cells (Figure 5E). These results indicated that FHL2 may accelerate the proliferation of bovine satellite cells.

### 3.6. FHL2 Knockdown Inhibited Differentiation of SMSCs

We also investigated the differentiation status of SMSCs by detecting the mRNA levels of four myogenic markers. The results showed that SMSCs transfected with FHL2 shRNA significantly decreased expression levels of MyoG, Myh3, Ckm and Mb (Figure 6A). Western blot found the protein level of MyHC and MyoG were decreased in FHL2 silenced cells (Figure 6B,C). Immunofluorescence of MyHC showed that FHL2 silencing significantly hindered the differentiation of SMSCs (Figure 6D,E). These data indicated that FHL2 silencing inhibits differentiation of the SMSCs into myotubes.

### 3.7. miR-377 Disrupted the Dvl-2-Mediated Wnt Signaling Pathway in Bovine SMSCs

To further study the regulatory mechanism of FHL2 in bovine SMSCs, RNA-seq found the expression of myogenesis-related genes was significantly decreased in FHL2 silenced cells (Figure 7A). KEGG pathway analysis indicated that the differentially expressed genes were clearly involved in the wnt signaling pathway, which is closely involved in muscle cell development (Figure 7B). We investigated its effects on four Wnt pathway target genes: c-Myc, wnt5a, wnt10b and lef1. FHL2 knockdown significantly inhibited the mRNA expression of all four of these genes (Figure 7C). The TOP/FOP assay found that the cells transfected with FHL2 shRNA significantly inhibited Wnt signaling activity than those transfected with control (Figure 7D). Furthermore, Western blot analysis FHL2 silencing significantly decreased the protein expression of Dvl-2, and also reduced the level of β-catenin in the nucleus (Figure 7E,F). In addition, we found that the cells transfected with miR-377 inhibitor significantly increased Dvl2 protein expression in both FHL2-silenced and control cells (Figure 7G), whereas overexpression of miR-377 significantly inhibited the expression of Dvl2 (Figure 7H). This suggests that miR-377 can affect Wnt/β-catenin signaling pathway activity by mediating Dvl2. We further explored the relationship between FHL2 and Dvl-2, using immunoprecipitation to detect a direct interaction between the two (Figure 7I). The results showed that miR-377 negatively controls the activity of the Wnt/β-catenin signaling by targeting FHL2.

## 4. Discussion

Many studies have reported that miR-377 affects the occurrence of tumors by regulating cell proliferation, migration and apoptosis. Nevertheless, the function of miR-377 in skeletal muscle growth and development has not been found. In this research, we found that miR-377 is highly abundant in bovine skeletal muscle and plays an essential role in regulating the proliferation and differentiation of bovine SMSCs. Wang et al. found that miR-377-3p restrains atherosclerosis-associated vascular smooth muscle cell proliferation and migration by targeting neuropilin2 [19]. Zhang also found that miR-377-3p restrains cell proliferation, migration and inflammation, and induces apoptosis of vascular smooth muscle cells by targeting CCND1, a key gene for cell proliferation [20]. Our study is consistent with previous studies and also found that miR-377 can inhibit the proliferation of bovine SMSCs. The principal effect of satellite cells in the growth of mature skeletal muscle is myonuclear accretion to support the transcriptional requirements of postnatal development [21]. Moreover, senescence leads to an inherent gene pathway disorder in satellite cells that affects their proliferation and differentiation in vivo and hinders muscle regeneration [22]. Moreover, we also found that miR-377 restrains the differentiation of SMSCs cells by inhibiting the expression of myogenic differentiation genes. These results manifest that miR-377 restrains the development of bovine skeletal muscle by inhibiting the proliferation and differentiation of SMSCs cells.

Most studies of miRNAs have concentrated on their ability to regulate cellular processes by targeting different genes. In our study, miR-377 was found to target FHL2 to regulate skeletal muscle development in bovine. FHL2 is primarily expressed in skeletal muscle and myocardium and has been proved to play an indispensable role in skeletal muscle development and structural maintenance [23]. Martin et al. found that FHL2 facilitates myoblast differentiation in mice by activating β-catenin [15]. Shi et al. found that FHL2 combines with Foxk1 and co-expresses Foxo4 activity in myogenic progenitors, enhancing muscle regeneration in adult skeletal muscle [24]. In this study, our results were consistent with previous reports showing that FHL2 plays a positive regulatory function in the proliferation and differentiation of bovine satellite cells.

The Wnt/β-catenin signaling pathway is critical in muscle development because Wnt ligands regulate the specification of paraxial mesoderm bone myoblasts and induce position-specific expression of myogenic regulatory factors [25]. Several articles have previously reported the effects of FHL2 on the Wnt signaling pathway [26,27,28]. Brun et al. found that FHL2 silencing inhibits tumor cell proliferation, and acts as an oncogene in osteosarcoma cells, decreasing Wnt signaling [26]. Yuan et al. reported that FHL2 is a co-activator of Wnt signaling in diabetic nephropathy and could play a key role in the treatment of diabetes [29]. In this research, we found that FHL2 interacts with Dvl2 to regulate Wnt/β-catenin signaling pathway and affects skeletal muscle development, which is a key component of the Wnt signaling pathway [30,31,32]. Moreover, we found miR-377 can inhibit the expression of Dvl2 and inhibit the Wnt/β-catenin signaling pathway. Huang et al. found that miR-377 inhibits colorectal cancer by negatively regulating the Wnt/β-catenin signaling pathway by targeting XIAP and ZEB2 [7]. Our study provides a new perspective for the negative regulation of Wnt/β-catenin signaling by miR-377. This suggests that miR-377 negatively regulates the Wnt/β-catenin signaling pathway by targeting FHL2, thereby inhibiting skeletal muscle growth and development.

Muscle generation is a strictly regulated process that begins with the generation of SMSCs, which are situated between the muscle cell membrane and the basal membrane, followed by activation and proliferation into myoblasts [3,4]. SMSCs remain resting in unstressed, mature muscle, until the muscle is damaged or stimulated, at which time they become synchronously activated to provide new myo-nuclei for the myofiber [33]. Mesenchymal stem cells also contribute to maintaining the integrity of muscle structure and function [34]. The growth rate of muscles during postpartum development is particularly important in animal meat due to their high economic value. Therefore, it is of great significance to explore new factors and mechanisms that regulate skeletal muscle development and physiology to improve human health and animal productivity. We identified miR-377 as a new regulatory factor that plays an important role in skeletal muscle development, which can provide a new theoretical basis for biological breeding and the prevention and treatment of skeletal muscle diseases.

## 5. Conclusions

In summary, our study shows that miR-377 plays an important role in bovine SMSCs proliferation and differentiation by negatively regulating the Wnt/β-catenin signaling pathway by targeting FHL2. We also found that the silencing in vitro of FHL2 and, consequently, the associated miR-377 that downregulate this expression in vivo, determines an inhibition of proliferation and differentiation of SMSCs in myotubes. miR-377 is involved in a variety of pathological processes of life activities, and a deeper understanding of its molecular regulatory mechanism is of great significance for the development of skeletal muscle and the prevention and treatment of muscle diseases.

## Figures and Tables

**Figure 1 genes-13-00947-f001:**
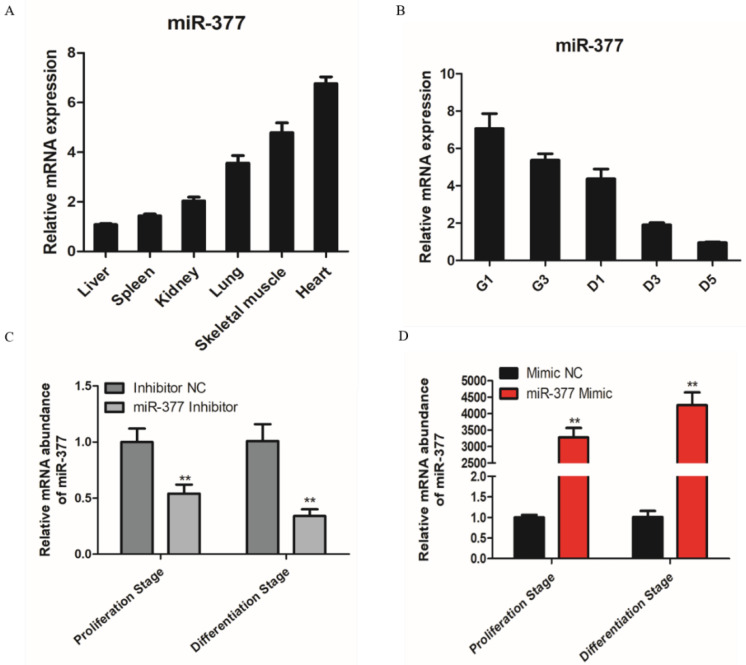
Expression pattern of miR-377 in bovine. (**A**) Expression levels of miR-377 in bovine heart, liver, spleen, lung, kidney and skeletal muscle. (**B**) Expression levels of miR-377 in SMSCs cultured in either GM for 1d and 3d (G1 and G3) or in DM for 1, 3 and 5d (D1, D3 and D5). (**C**) The expression levels of miR-377 in bovine SMSCs transfected with inhibitor NC and miR-377 inhibitor. (**D**) The expression levels of miR-377 in bovine SMSCs transfected with Mimic NC and miR-377 Mimic. Data are expressed as mean ± SEM (*n* = 3). ** *p* < 0.01.

**Figure 2 genes-13-00947-f002:**
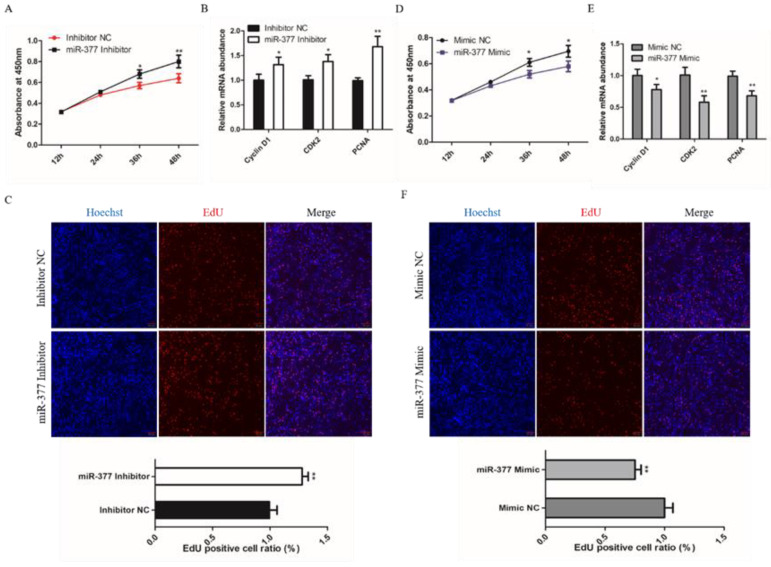
Effect of miR-377 on the proliferation of bovine SMSCs. (**A**) CCK8 analysis equal numbers of bovine SMSCs transfected with inhibitor NC and miR-377 inhibitor in GM for 12, 24, 36 and 48 h. The absorbance at 450 nm on the Y-axis indicates the number of viable cells. (**B**) qPCR analysis the expression levels of CyclinD1, CDK2 and PCNA in inhibitor NC and miR-377 inhibitor cells. (**C**) EdU staining of bovine SMSCs after transfected with inhibitor NC and miR-377 inhibitor. The lower histogram represents the proliferation rate. (**D**) CCK8 analysis equal numbers of bovine SMSCs transfected with Mimic NC and miR-377 Mimic in GM for 12, 24, 36 and 48 h. The absorbance at 450 nm on the Y-axis indicates the number of viable cells. (**E**) qPCR analysis the expression levels of CyclinD1, CDK2 and PCNA in Mimic NC and miR-377 Mimic cells. (**F**) EdU staining of bovine SMSCs after transfected with Mimic NC and miR-377 Mimic. The lower histogram represents the proliferation rate. Data are expressed as mean ± SEM (*n* = 3). * *p* < 0.05; ** *p* < 0.01.

**Figure 3 genes-13-00947-f003:**
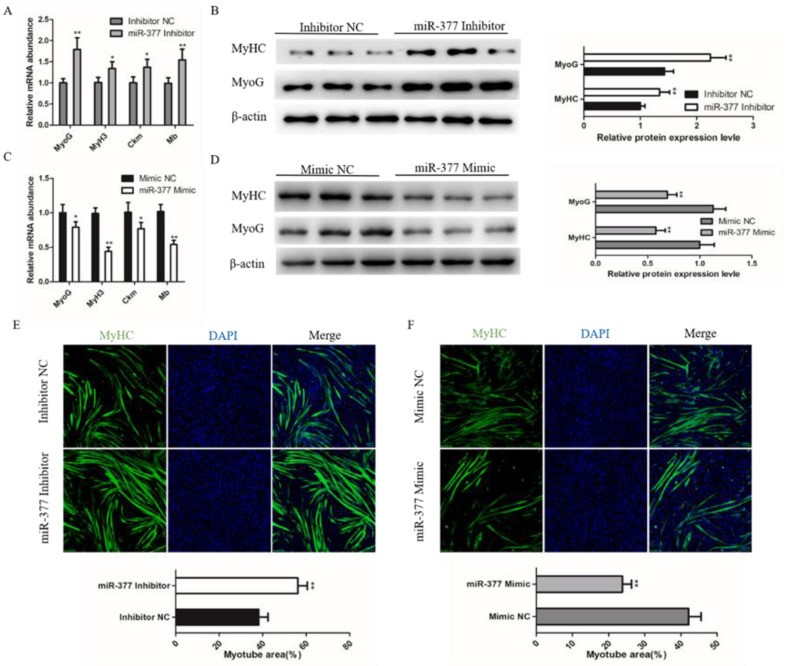
Effect of miR-377 on the differentiation of bovine SMSCs. (**A**) qPCR analysis the expression levels of MyoG, MyH3, Ckm and Mb in inhibitor NC and miR-377 inhibitor cells. (**B**) Western blot analysis the protein levels of MyoG and MyHC in inhibitor NC and miR-377 inhibitor cells. (**C**) qPCR analysis the expression levles of MyoG, MyH3, Ckm and Mb in Mimic NC and miR-377 Mimic cells. (**D**) Western blot analysis the protein levels of MyoG and MyHC in Mimic NC and miR-377 Mimic cells. (**E**) SMSCs transfected with inhibitor NC and miR-377 inhibitor were induced differentiate for 72 h then stained with MyHC antibody and DAPI. The lower histogram represents the myotube area. (**F**) SMSCs transfected with Mimic NC and miR-377 Mimic were induced differentiate for 72 h then stained with MyHC antibody and DAPI. The lower histogram represents the myotube area. Data are expressed as mean ± SEM (*n* = 3). * *p* < 0.05; ** *p* < 0.01.

**Figure 4 genes-13-00947-f004:**
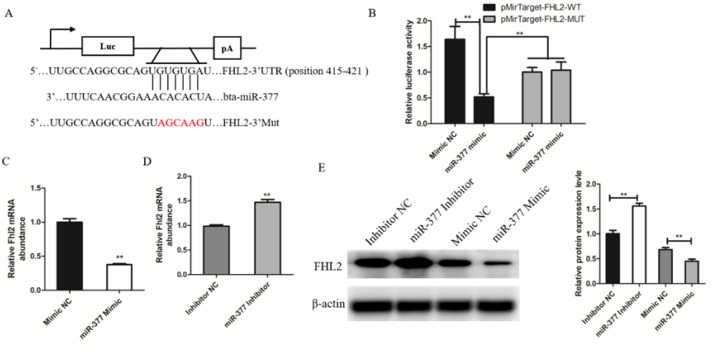
FHL2 is a target gene of miR-337. (**A**) The target gene of miR-37 was predicted by TargetScan. (**B**) The dual-luciferase reporter analysis showing that FHL2 was a direct target of miR-377. (**C**) The mRNA expression levels of FHL2 in Mimic NC or miR-377 Mimic cells by qPCR. (**D**) The mRNA expression levels of FHL2 in Inhibitor NC or miR-377 Inhibitor cells by qPCR. (**E**) Western blot analysis the protein expression levels of FHL2 after transfected with Mimic NC, miR-377 Mimic, Inhibitor NC or miR-377 Inhibitor. Data are expressed as mean ± SEM (*n* = 3). ** *p* < 0.01.

**Figure 5 genes-13-00947-f005:**
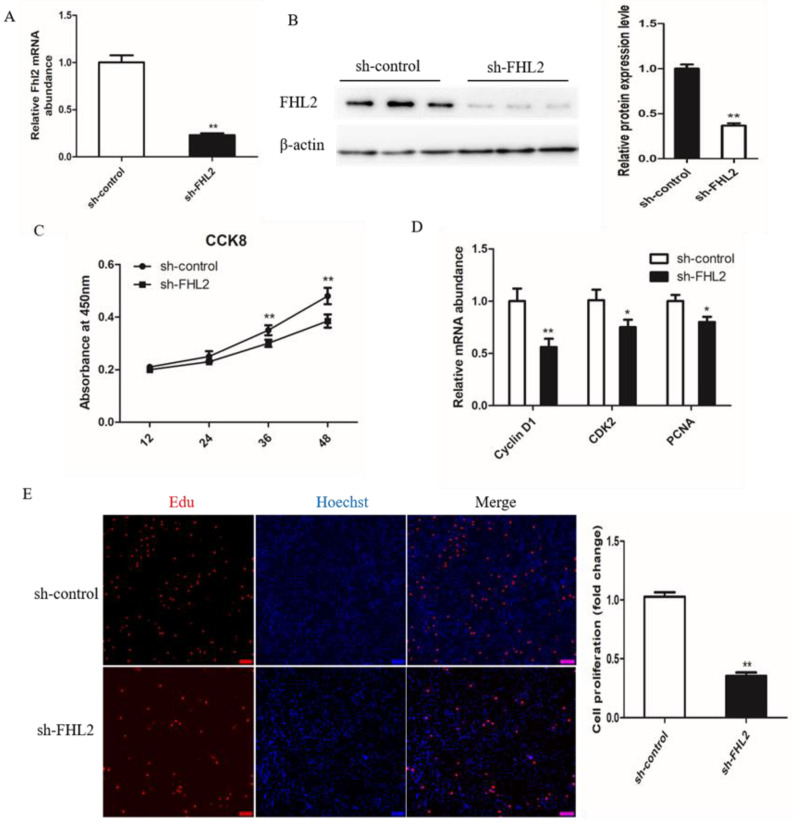
Effect of FHL2 knockdown on proliferation of bovine SMSCs. (**A**) Relative expression levels of FHL2 mRNA in FHL2 silenced and control cells. (**B**) Western blot analysis the protein levels of FHL2 in FHL2 silenced and control cells. (**C**) CCK8 analysis equal numbers of bovine SMSCs transfected with FHL2 shRNA and control in GM for 12, 24, 36 and 48 h. (**D**) qPCR analysis the expression levels of CyclinD1, CDK2 and PCNA in FHL2 silenced and control cells. (**E**) EdU staining of bovine SMSCs after transfected with FHL2 shRNA and control. Data are expressed as mean ± SEM (*n* = 3). * *p* < 0.05; ** *p* < 0.01.

**Figure 6 genes-13-00947-f006:**
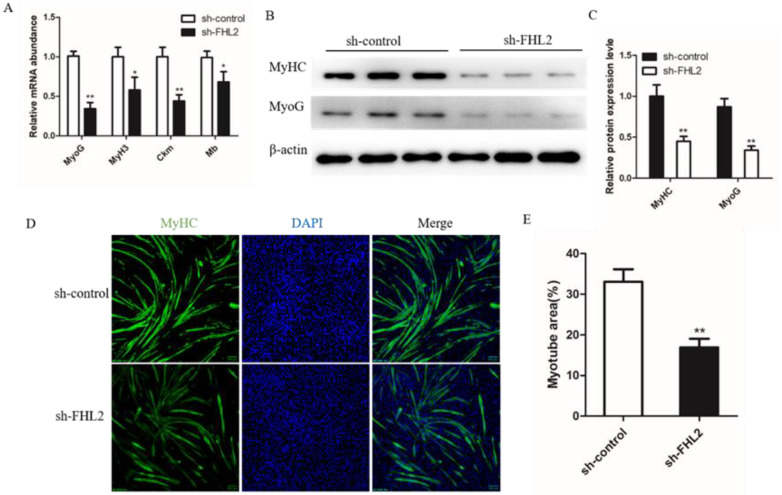
Effect of FHL2 knockdown on differentiation of bovine satellite cells. (**A**) Relative mRNA expression levels of MyoG, MyH3, Ckm and Mb in FHL2 silenced and sh-control cells. (**B**,**C**) The protein expression levels of MyHC and MyoG in FHL2 silenced and sh-control cells by Western blot assay. (**D**) Immunofluorescence staining for MyHC after transfected with FHL2 shRNA and sh-control cells. (**E**) Calculated percentage of myotube area in FHL2 silenced and sh-control cells. Data are expressed as mean ± SEM (*n* = 3). * *p* < 0.05; ** *p* < 0.01.

**Figure 7 genes-13-00947-f007:**
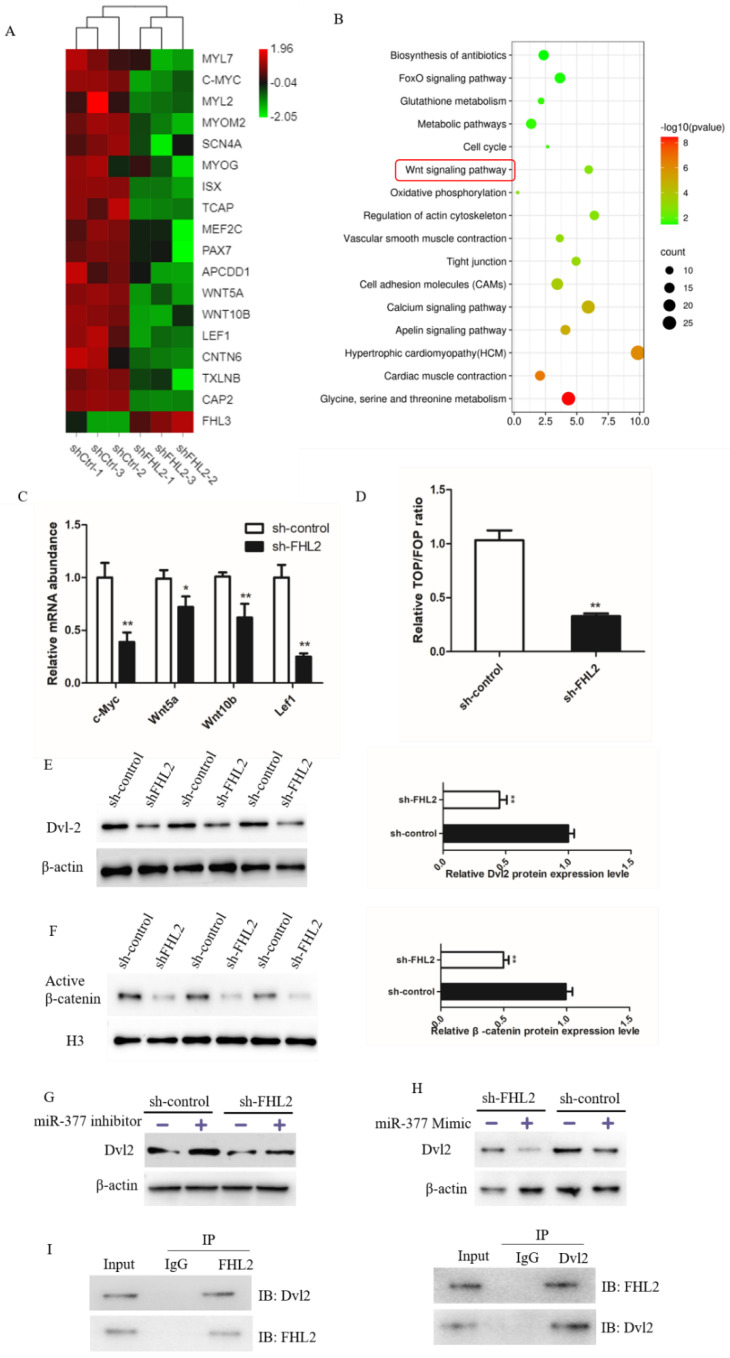
miR-377 negatively regulates Wnt/β-catenin signaling pathway by mediating Dvl-2. (**A**) Grading clustering and heat map of gene expression levels associated with marker myogenesis in SMSCs transfected with control or FHL2 shRNA. (**B**) Enrichment analysis of pathways with significantly differentially expressed genes in sh-ctrl and FHL2 silenced cells. (**C**) Relative mRNA expression levels of c-Myc, Wnt5a, Wnt10b and Lef1 in FHL2 silenced and sh-control cells. (**D**) Luciferase activity of TOP/FOP in control and FHL2 silenced cells. (**E**) Western blot analysis Dvl-2 protein levels in FHL2 silenced and sh-control cells. (**F**) Western blot analysis of β-catenin protein expression in the nucleus of sh-control or FHL2 silenced cell. (**G**) Western blot analysis Dvl2 protein expression levels in FHL2 silenced and sh-control cells treated with miR-377 inhibitor for 24 h. (**H**) Western blot analysis of Dvl2 protein expression levels in FHL2 silenced and control cells treated with miR-377 Mimic for 24 h. (**I**) Co-immunoprecipitation analysis of FHL2 and Dvl2 after SMSCs were cultured in differentiation medium for 72 h. Data are expressed as mean ± SEM (*n* = 3). * *p* < 0.05; ** *p* < 0.01.

**Table 1 genes-13-00947-t001:** RNA oligonucleotides in this study.

Name	Forward Primer (5′–3′)
FHL2-shRNA	GCAAGGACUUGUCUUAUAATT
	UUAUAAGACAAGUCCUUGCTT
MiR-377 mimic	AUCACACAAAGGCAACUUUUGU
Mimic NC	UUGUACUACACAAAAGUACUG
MiR-377 inhibitor	ACAAAAGUUGCCUUUGUGUGAU
Inhibitor NC	CAGUACUUUUGUGUAGUACAA

**Table 2 genes-13-00947-t002:** Primer information for quantitative real-time PCR analysis.

Genes	Forward Primer (5′–3′)	Size (bp)	Annealing Temperature (°C)	Accession Number
*FHL2*	F: CTCTGCGCTTCTCAGCGATA	128	61	NM_001046046.2
	R: GGCAGGAAGTTACACCGGAA			
*Ckm*	F: CCTGACGGGTGAGTTCAAGG	187	60	NM_174773.4
	R: TGATCCTCCTCGTTCACCCA			
*MyoG*	F: TGGGCGTGTAAGGTGTGTAA	78	60	NM_001111325.1
	R: TATGGGAGCTGCATTCACTG			
*MyH7*	F: CTTCAACCACCACATGTTCG	178	58	NM_174727.1
	R: GCTTCTGGAAGTTGCTGGAC			
*Wnt5a*	F: CAACTGGCAGGACTTTCTCAA	127	61	NM_174727.1
	R: CATCTCCGATGCCGGAACT			
*Wnt10b*	F: CTCTGCCACAGCCAAACTCT	106	60	XM_010805029.3
	R: ATCGAACTTGCCTGGCTTGA			
*c-Myc*	F: GTAATTCCAGCGAGAGGCAGA	213	60	NM_001046074.2
	R: CTAGGCTAGCTCGGCTCTTC			
*Lef1*	F: CCCTGTGTTGTTCGGCCTC	271	59	NM_001192856.1
	R: ATTGGAAGGATGCGTCAGGG			
*CyclinD1*	F: ATGAAGGAGACCATCCCCCT	117	61	NM_001046273.2
	R: CGCCAGGTTCCACTTGAGTT			
*CDK2*	F: AGGGAACGTACGGAGTTGTG	78	58	NM_001014934.1
	R: GACATCCAGCAGCTTGACAAT			
*PCNA*	F: TCCAGAACAAGAGTATAGC	162	60	NM_001034494.1
	R: TACAACAGCATCTCCAAT			
*miR-377*	F: ATCACACAAAGGCAACTTTTGT	/	60	/
	R: CAGGTCCAGTTTTTTTTTTTTTT			
*MB*	F: ACTGACCTGCACCTTTACCC	210	62	NM_173881.2
	R: CTCAGGGCAAGCAAGACACT			
*β-actin*	F: CATCCTGACCCTCAAGTA	146	56–62	NM_173979.3
	R: CTCGTTGTAGAAGGTGTG			
*U6*	F: GCTTCGGCAGCACATATACTAAAAT	/	60	/
	R: CGCTTCACGAATTTGCGTGTCAT			

## Data Availability

All data generated or analyzed during this study are included in this published article.

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
