# Peer review of "miR-377 Inhibits Proliferation and Differentiation of Bovine Skeletal Muscle Satellite Cells by Targeting FHL2"

_genes, 2022, doi:10.3390/genes13060947_

Round 1
Reviewer 1 Report
Review Report “Genes”
MiR-377 inhibit proliferation and differentiation through negative regulation wnt/β-catenin signaling by targeting FHL2 of bovine skeletal muscle satellite cells
Brief summary:
This manuscript focuses on the study of MiR-377 as inhibitor of proliferation and differentiation of bovine skeletal muscle satellite cells by targeting FHL2 gene and down-regulating Wnt/β-catenin signaling pathway. Non-coding RNAs (ncRNAs) regulate the proliferation and the differentiation of muscle satellite cells SMSCs, that plays an important role in myogenesis. Skeletal muscle is the most important component of the carcass of an animal raised for meat, and accounts for 45%-60% of its body weight at the adult stage. Furthermore, it is involved in maintaining of body’s movement, respiration and metabolism.
Broad comments:
This manuscript focuses on the study of MiR-377 as inhibitor of proliferation and differentiation of bovine skeletal muscle satellite cells by targeting FHL2 gene and down-regulating Wnt/β-catenin signaling pathway. Non-coding RNAs (ncRNAs) regulate the proliferation and the differentiation of muscle satellite cells SMSCs, that plays an important role in myogenesis. Skeletal muscle is the most important component of the carcass of an animal raised for meat, and accounts for 45%-60% of its body weight at the adult stage. Furthermore, it is involved in maintaining of body’s movement, respiration and metabolism. The experimental design is good as well as the obtained results, that are promising in order to clarify some aspects of this field, in particular for prevention and treatment of muscle disease in bovine and other mammalian. In this way, I suggest to implement the introduction with recent references about SMSCs and mi-RNA (miR-377) role and the statistical analysis for which I would reserve a paragraph, if possible.
Please, correct some grammar and orthographical mistakes along the text, revising English.
Some specific comments:
Title:
I suggest to change the title for example into: MiR-377 inhibits proliferation and differentiation of bovine skeletal muscle satellite cells by targeting FHL2. At the most I’d add “down-regulating the wnt/β-catenin signaling pathway. There is also a grammar mistake.
Introduction
I suggest to implement the introduction with recent references about SMSCs and mi-RNA (miR-377) role.
Material and methods
2.3 paragraph
Since real-time PCR was performed, please specify if SYBR-green or Taqman assay was used. I can image the first assay, because sequence probe is not reported, but I suggest to report in the text, independently from the references and protocols to which you refer.
Line 71: please remove a % symbol, because is double-reported.
Line 76: please convert 105 in 105
Line 99: please define PBS acronym
Line 105: please add and define SDS acronym
Results
Line 167: better specify and describe the target assay (qpCR, EDU, CCK8).
Line 223: please remove d from “confirmed” and “these”, so “to confirm” or better “in order to confirm this result
Line 268: specifiy that “FHL2 silencing inhibits the proliferation and the differentiation of the SMSCs into myotubes”.
Discussion and Conclusions
I suggest to specify also in “conclusions” section that the silencing in vitro of FHL2 and consequently the associated miR-377 that down-regulate his expression in vivo, determines an inhibition of proliferation and differentiation of SMSCs in myotubes.
References
Better check the reported references also in accordance with the format required by “Genes-MDPI”.
Author Response
Dear editor and reviewers:
Thank you very much for your comments concerning our manuscript, your comments are all valuable and very helpful for revising and improving our paper, as well as the important guiding significance to our researches. We have studied comments carefully and have made correction. The revised portion can be tracked in the paper by red color. I hope this revision can make my paper more acceptable. The main corrections in the paper and the responds to your comments are addressed point by point below.
- I suggest to change the title for example into: MiR-377 inhibits proliferation and differentiation of bovine skeletal muscle satellite cells by targeting FHL2. At the most I’d add “down-regulating the wnt/β-catenin signaling pathway. There is also a grammar mistake.
Response: Thanks for your suggestion, I have made revised in the manuscript.
- I suggest to implement the introduction with recent references about SMSCs and mi-RNA (miR-377) role.
Response: Thank you for your comment. I agree with you very much, but there is no report on miR-377 on SMSCs so far.
- Since real-time PCR was performed, please specify if SYBR-green or Taqman assay was used. I can image the first assay, because sequence probe is not reported, but I suggest to report in the text, independently from the references and protocols to which you refer.
Response: Thank you for your suggestion. I have made a supplement in the manuscript in line 95.
- Line 71: please remove a % symbol, because is double-reported.
Response: Thank you for your suggestion. We have removed it.
- Line 76: please convert 105 in 105
Response: I'm sorry for this mistake, and I've corrected it.
- Line 99: please define PBS acronym
Response: Thank you for your suggestion. I have defined it.
- Line 105: please add and define SDS acronym
Response: Thank you for your suggestion. I have defined it.
- Line 167: better specify and describe the target assay (qpCR, EDU, CCK8).
Response: Thank you for your suggestion. I have described it.
- Line 223: please remove d from “confirmed” and “these”, so “to confirm” or better “in order to confirm this result
Response: Thank you for your suggestion. I have revised it.
- Line 268: specifiy that “FHL2 silencing inhibits the proliferation and the differentiation of the SMSCs into myotubes”.
Response: Thank you for your suggestion. I have revised it.
- I suggest to specify also in “conclusions” section that the silencing in vitro of FHL2 and consequently the associated miR-377 that down-regulate his expression in vivo, determines an inhibition of proliferation and differentiation of SMSCs in myotubes.
Response: Thank you for your suggestion. I have revised it.
- Better check the reported references also in accordance with the format required by “Genes-MDPI”.
Response: Thank you for your suggestion. I have revised it.

Reviewer 2 Report
Introduction
This is OK and it sets the scene well.
However, I suggest that the authors expand more the third paragraph to provide a deeper insight for future readers.
Procedures
2.2. We were not fully satisfied with Lipofectamine. Can the authors provide some more detailed data in a supplementary table to confirm that the material run well?
All other reagents are OK.
Table 2. Please expand by including all the details for running the PCR (temperature, product size etc.) Also, the revised table should be moved to supplementary table.
2.9. How did you deal with technical repetitions within experiments? Please describe.
Results
I suggest to increase the size of graphs, as details are difficult to read.
Discussion
This is a weak part of the manuscript and must be revised carefully.
For such an extensive experiment, the discussion is shallow and does not cover fully all the points and facets of the results.
I strongly suggest to expand the discussion, by going into greater depth into the findings and by presenting alternative hypotheses.
Overall. The manuscript can advance to the next stage, but significant and extensive revision is needed before reconsideration.
Author Response
Dear editor and reviewers:
Thank you very much for your comments concerning our manuscript, your comments are all valuable and very helpful for revising and improving our paper, as well as the important guiding significance to our researches. We have studied comments carefully and have made correction. The revised portion can be tracked in the paper by red color. I hope this revision can make my paper more acceptable. The main corrections in the paper and the responds to your comments are addressed point by point below.
- However, I suggest that the authors expand more the third paragraph to provide a deeper insight for future readers.
Response: Thank you for your suggestion. I have revised it.
- We were not fully satisfied with Lipofectamine. Can the authors provide some more detailed data in a supplementary table to confirm that the material run well?
Response: Thanks for your comment, Lipofectamine is a very common transfection reagent with a large number of articles published. We detected the mRNA and protein expression of the target gene after transfection and found that the transfection effect was good.
- Table 2. Please expand by including all the details for running the PCR (temperature, product size etc.) Also, the revised table should be moved to supplementary table.
Response: Thank you for your suggestion. I've added it.
- How did you deal with technical repetitions within experiments? Please describe.
Response: Thank you for your suggestion. All experiments included at least three biological replicates. I've described it.
- I suggest to increase the size of graphs, as details are difficult to read.
Response: Thank you for your suggestion. I have increased it.
- I strongly suggest to expand the discussion, by going into greater depth into the findings and by presenting alternative hypotheses.
Response: Thank you for your suggestion. We have expanded our discussion according to your request.

Round 2
Reviewer 2 Report
In general, the authors revised the manuscript well. One point however was not dealt with correctly.
The authors must describe in detail the data management and statistical analysis performed in the results of the three technical replicates performed for each sample.
After that, the manuscript can be accepted.
Author Response
The authors must describe in detail the data management and statistical analysis performed in the results of the three technical replicates performed for each sample.
Response: Thank you for your suggestion, I have revised in line 144.